# APPA: AGENTIC PREFORMULATION PATHWAY ASSISTANT

**Julius Lange, Leonid Komissarov, Nicole Wyttenbach & Andrea Anelli***
Roche Pharmaceutical Research and Early Development
F. Hoffmann-La Roche Ltd.
Basel, Switzerland
`andrea.anelli@roche.com`

## ABSTRACT

The design and development of effective drug formulations is a critical process in pharmaceutical research, particularly for small molecule active pharmaceutical ingredients. This paper introduces a novel agentic preformulation pathway assistant (APPA), leveraging large language models coupled to experimental databases and a suite of machine learning models to streamline the preformulation process of drug candidates. APPA successfully integrates domain expertise from scientific publications, databases holding experimental results, and machine learning predictors to reason and propose optimal preformulation strategies based on the current evidence. This results in case-specific user guidance for the developability assessment of a new drug and directs towards the most promising experimental route, significantly reducing the time and resources required for the manual collection and analysis of existing evidence. The approach aims to accelerate the transition of promising compounds from discovery to preclinical and clinical testing.

## 1 INTRODUCTION

The journey of a new drug candidate from discovery to clinical trials is beset by numerous challenges, including staggering R&D costs and high attrition rates (Kola & Landis, 2004; Simoens & Huys, 2021). Many promising small-molecule active pharmaceutical ingredients (APIs) suffer from unfavorable physicochemical properties, such as low solubility or permeability, which in turn reduce their bioavailability and efficacy (Lipinski et al., 1997; Amidon et al., 1995). In many cases, these limitations can be mitigated by using suitable formulation approaches. A substantial body of literature describes the various factors impacting API formulations and outlines potential strategies to address them (Zhang et al., 2020; Florence & Attwood, 2015; Nair et al., 2020). One notable example is the Developability Classification System (DCS) (Butler & Dressman, 2010), closely related to the Biopharmaceutics Classification System (BCS) (Amidon et al., 1995) and the Biopharmaceutics Drug Disposition Classification System (BDDCS) (Benet, 2013). These frameworks categorize drug candidates by their solubility and permeability profiles, offering an initial guide to identifying APIs with potential oral bioavailability issues, and assist in making informed decisions about clinical formulation and development strategies. Despite these systematic approaches, formulation remains time- and resource-intensive, requiring extensive experimentation, expert knowledge, and careful consideration of multiple physicochemical parameters (Yu, 2008).

Meanwhile, machine learning (ML) has emerged as a powerful enabler in drug development (Lange et al., 2024a; Komissarov et al., 2024; Atz et al., 2024), with demonstrated potential for accelerating tasks like solubility and permeability prediction and reducing the need for direct experimental validation (Sliwoski et al., 2014; Chen et al., 2018; Vamathevan et al., 2019; Lange et al., 2024b; Hornick et al., 2024). Nevertheless, formulation scientists must still weave together disparate data streams, ranging from experimental measurements to predictive modeling, best practice documents, and laboratory standard operating procedures (SOPs) into an effective "first-time-right" formulation strategy. Managing such diverse information is time-consuming and prone to human error, especially at a stage where efficiency and speed are paramount.

To address these challenges, we introduce APPA, an agentic workflow designed to guide preformulation design. APPA integrates established ML models for solubility and permeability prediction with the reasoning capabilities of a large language model (LLM), enabling automated analysis of limited (experimental) data — such as an API's structure and initial solubility measurements — while providing case-specific recommendations informed by the broader scientific literature. By unifying key decision-making processes and suggesting the most promising experimental pathways, APPA aims to substantially reduce the time, costs, and risks associated with preformulation in drug discovery.

## 2 APPA

APPA is an LLM-based agent powered by GPT-4o (OpenAI et al., 2024), designed to guide preformulation workflows by analyzing and predicting drug candidate properties. A typical process begins with the user providing at least one compound identifier, which APPA uses to retrieve the corresponding chemical structure of an API. From there, the user can query the agent about existing experimental results, request comparisons between multiple compounds, or investigate specific questions, such as proposing a suitable formulation approach for a specific oral drug dose.

In this work, we center our approach on deriving the Developability Classification System (DCS) class of a drug candidate, but the same workflow can be easily adapted to other classification frameworks (*e.g.* , BCS or preclinical dose number (PDo) (Wuelfing et al., 2015)). Depending on the resulting classification, APPA recommends the most promising next steps in the experimental cascade, supporting these suggestions with quantitative reasoning. A high-level example of the APPA workflow is shown in Figure 1.

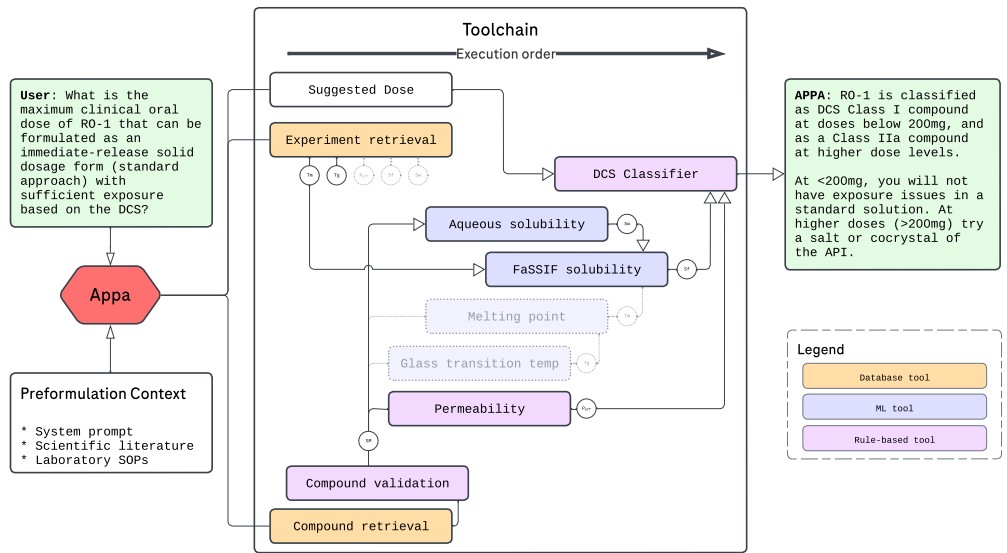

Figure 1: Example diagram of an APPA workflow. A user can make a request to the agent regarding their compound of interest (here presented as an internal code-name). The tool will consult its system prompt and Laboratory's SOPs to define the perimeter of the answer. Depending on the nature of the request, a chain of tools will be called to achieve the desired answer - here shown in full color. Outputs from tools that are depicted early in the execution order (XO) can serve as inputs for interconnected tools depicted later in the XO. APPA uses pretrained ML models based on internal experimental data and empirical models to impute relevant preformulation parameters. The output of the query is a structured formulation recommendation.

To achieve this functionality, APPA leverages the Langchain framework (Chase, 2022) and is equipped with a set of specialized "tools." We categorize these tools into two main groups:

1. **Databases**, which retrieve the chemical structure and any available experimental data.

2. **Calculators**, which require computational resources to produce an output. This includes data-driven (ML-based) and empirical (rule-based) models.

In addition, we provide relevant scientific context via an appropriate system prompt and retrieval-augmented generation (Lewis et al., 2020) of laboratory standard operating procedures (see Appendix A.2 for more details).

When a user query cannot be answered solely by referencing experimental data, APPA seamlessly invokes its predictive tools to estimate or impute missing information. Specifically, we integrate models capable of predicting a wide range of physicochemical properties and assay outcomes, including melting point, glass transition temperature, aqueous solubility (Lovrić et al., 2021), solubility in fasted state simulated intestinal fluid (FaSSIF) (Dressman et al., 1998), human intestinal permeability ($P_{eff}$), and DCS class. The machine learning models deployed here implement largely the models presented by Lange et al. (2024b).

Notably, APPA can chain the outputs of multiple tools to address more complex questions. For instance, estimating the solubility of a compound in FaSSIF may require combining its chemical structure with predicted aqueous solubility and melting point. By orchestrating these predictive and database tools in a single agentic interface, APPA provides a streamlined path toward more efficient and informed decision-making in the preformulation stage.

## 3 RESULTS

We evaluate APPA against its ability to accurately predict the class of a compound based on the Developability Classification System (DCS) (Butler & Dressman, 2010). In a first quantitative evaluation we generate 500 virtual data points by randomly selecting drug-like compounds from the ChEMBL database Zdrazil et al. (2024) and assign each a melting point and dose in the range of $(80, 400)\,°\mathrm{C}$ and $(5, 800)\,\mathrm{mg}$, respectively. For each point we manually compute DCS class and use it as the ground truth when evaluating the performance of an LLM-based application. We compare APPA to a non-agentic instance of GPT-4o where the relevant literature for the DCS classification has been made available through a context. The following prompt is used to generate the responses:

```
User: What is the DCS class of SMILES [SMILES] at a dose of [DOSE] mg and a melting point of [
    MELTINGPOINT] degC?
```

Where the items in square brackets symbolize variable user input. This purely theoretical exercise aims to establish a baseline LLM-performance, the ability of an agent to understand and use the provided tools in the correct order, as well as test for potential hallucinations. We report the results, expressed through the balanced accuracy and F1-micro scores in Table 1.

Table 1: Balanced accuracy and F1-micro scores on the ChEMBL data set of virtual compounds comparing APPA to a baseline GPT-4o instance.

| Method | Accuracy | F1-micro |
|---|---|---|
| APPA | 0.895 | 0.938 |
| GPT-4o + Context | 0.156 | 0.186 |

The poor performance of the GPT-4o instance, which is likely due to a lack of chemical understanding, highlights the importance of setting where domain knowledge is required.

As a second exercise, in a more end-to-end fashion, we evaluate APPA by asking it for the the DCS class of a number of marketed drugs and check whether it is in line with the results published by Butler & Dressman (2010). The same prompt as above is used, however, SMILES is replaced with an internal compound ID when querying APPA in order to additionally test the retrieval of data from databases. We report the results in Table 2.

Not only does APPA correctly classify all four example compounds, but also provides meaningful next steps with regards to the formulation of the compound with a given class. As an example of followup suggestion, APPA could recommend a specific choice of molecular vehicle to facilitate the preformulation task, based on the available tools. For example, in presence of a class IIb compound,

Table 2: DCS Classification of four marketed drugs with APPA, GPT-4o, and their reference class as reported by Butler & Dressman (2010). APPA additionally provides suggestions for the next viable formulation steps based on the predicted class. Dose and melting point ($T_m$) reported in mg and °C, respectively.

| Drug | Dose | $T_m$ | Ref. | APPA | GPT-4o | APPA Suggestions |
|---|---|---|---|---|---|---|
| Paracetamol | 500 | 170 | I | I | IIa | a |
| Nitrendipine | 80 | 158 | IIa | IIa | IIb | b, c |
| Griseofulvin | 500 | 220 | IIb | IIb | IIb | d, e, f |
| Aciclovir | 400 | 257 | III | III | IV | a, g, h |

[a] Immediate release dosage form, [b] Particle-size reduction
[c] Co-crystallization or salt formation [d] pH modification (if ionisable), [e] Complexation
[f] Amorphous solid dispersions, [g] Prodrugs, [h] Mucoadhesive delivery system

a lipid based formulation could be followed. APPA in this case would aptly call the ML models to predict solubility in medium-chain triglyceride and olive oil (which are implemented in the current version) and compare their theoretical performances to recommend which vehicle to try first. This example is further shown in A.3. The accuracy of the results rests on the quantitative evidence integrated in the tools: The model can defend its recommendations with numerical evidence - as shown in Section A.4 of the Appendix. We furthermore report that by casting the available tools into an agentic framework, users are no longer bound by a potentially limiting static user interface. As such, we can report additional successful experiments that would allow for an easy comparison of multiple compounds at once, or a parameter sweep in search of the optimal conditions (*e.g.* finding the maximum dose while staying DCS Class I or III). We can for example summon a direct comparison of aqueous solubility between multiple compounds:

```
User: Which one is more soluble in water: RO-1 or RO-2?
```

```
Appa: The compound RO-1 is more soluble in water with an aqueous solubility of 10.35 mg/mL
    compared to RO-2 which has an aqueous solubility of 0.61 mg/mL.
```

A user can ask which maximum dose we could achieve without complicating the formulation strategy for a given compound:

```
User: Calculate what is the maximal dose in mg that compound RO-3 can be administered while
    still classify as a DCS class 1. Be precise to the single digit. Assume a melting
    temperature of of 170 degC. Explain why.
```

```
Appa: The maximal dose in mg that compound RO-3 can be administered while still classifying as
    a DCS class 1 is 23 mg. The dose number for a 1 mg dose was found to be 0.043, which
    means the dose can be increased to approximately 23 mg to still be within the DCS class 1
    criteria.
```

Notably, APPA will automatically combine the tools at hand to adapt to the user queries, making the design of a static user interface with hard-coded workflows redundant.

## 4 DISCUSSION

Initial results from integrating access to experimental data with predictive models through an LLM agent are highly promising. In particular, we do realize the potential of such agents as one-stop shops that can query and meaningfully combine various otherwise fragmented tools. Reducing the need for context switching and manual integration of data from multiple sources is especially impactful in the pharmaceutical industry, where the need to develop drugs efficiently is key. Agents are also attractive from a developer's or data scientist's perspective, as fewer individual workflows need to be written manually. On the other hand, issues typical to large language models, such as hallucination (Huang et al., 2023) need to be properly addressed before LLM agents can be

fully adapted by the scientific community. For example, an area where APPA currently struggles is the detection of missing input, where instead of stopping the reasoning chain and asking for user input, the agent would proceed with 'invented' values. Particularly in the natural sciences, where domains of expertise can be very deep, building user trust in automated workflows can be challenging. For this reason we are considering a human-in-the-loop approach where the agent would list the individual tools it would like to use at each step and ask for user confirmation before proceeding. Large research organizations will naturally produce a variety of digital tools – we believe that once the above issues related to LLM agents are addressed, there is ample potential for them to become a generic interface, promoting integration and ease of access and multi-modal reasoning.

## 5 CONCLUSION

APPA represents a viable solution to a fragmented tool and information landscape, common in scientific discovery. By meaningfully combining predictive machine learning models and database access with large language models, APPA provides a novel, easily-accessible approach to the developability assessment of new drug candidates and preformulation design, which has the potential to significantly accelerate the development of new drugs. The use of agentic frameworks enables the creation of a flexible and adaptable workflows, which can be easily customized to different APIs and experimental settings. Future work will be focusing on extending APPA to act on the information provided by the DCS classification: Various additional data and models can be integrated to all following up on the specific formulation suggestions, further highlighting the most promising route to pursue. Moreover, additional as validation of the performance on a wider range of APIs is ongoing.

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

# A  APPENDIX

## A.1  TOOLS

APPA has access to the following tools, categorized by their level of complexity and dependencies:

**Level 0: Basic Compound Information Retrieval**

- `code_to_smiles(code)`: Converts a compound code to its corresponding SMILES string.
- `fetch_exp_data(code)`: Retrieves physicochemical data for a given compound code from a database API.
- `is_smiles_valid(smiles)`: Checks the validity of a given SMILES string. Crucial for ensuring subsequent tools operate on correct molecular representations.

**Level 1: Developability Classification System (DCS)**

- `get_dcs_class(sol_fassif, peff, dose)`: Classifies a compound according to the DCS based on its solubility in FaSSIF, effective permeability (peff), and dose.

**Level 2: Key Property Predictions**

- `calculate_fassif_solubility(smiles, melting_point, sol_buffered)`: Predicts solubility in FaSSIF. Requires melting point and aqueous solubility.
- `calculate_peff(smiles)`: Calculates human effective permeability (peff) from a SMILES string.

**Level 3: Advanced Solubility Predictions** These predictive tools are based on machine learning models built on internal data. They all combine cheminformatic fingerprints with additional experimental input (e.g. $T_m$, $T_g$, $S_{buff}$ ..etc) and 3-D structural information.

- `calculate_melting_point(smiles, T_g)`: Predicts melting point. Requires the glass transition temperature ($T_g$).
- `calculate_aqueous_solubility(smiles)`: Predicts aqueous solubility.
- `calculate_sol_in_mct(smiles, T_m)`: Predicts solubility in medium chain triglycerides (MCT). Requires melting point ($T_m$).
- `calculate_sol_in_olive_oil(smiles, T_m)`: Predicts solubility in olive oil. Requires melting point ($T_m$).
- `calculate_sol_in_propylene_glycol(smiles, T_m)`: Predicts solubility in propylene glycol. Requires melting point ($T_m$).
- `calculate_sol_in_polysorbate(smiles, T_m, sol_buffered)`: Predicts solubility in Polysorbate 80. Requires melting point ($T_m$) and buffered solubility.
- `calculate_sol_in_cyclodextrine(smiles, T_m, sol_buffered)`: Predicts solubility in cyclodextrines. Requires melting point ($T_m$) and buffered solubility.

## A.2   SYSTEM PROMPTS

The system prompt we used for our example contains a description of the workflow our agent uses. The basic system prompt is as follows:

```
You are a virtual formulation assistent with several tools at hand.
You always answer questions using these tools.
Carefully check the following at every step:
If you are missing input data to use a tool you stop the reasoning process and inform the
    human about the missing data.
If you can not answer a question you do not hypothesize but inform the human about the missing
     data.
Your final answer will include a list of tools that you used to arrive at the conclusion and a
     short reason why the tool was used.
Ensure the above rules are followed strictly.

After classifying it with DCS, always try to use the correct followup tools to suggest a
    specific strategy before returning the final result.
```

The system prompt is used to provide the agent a minimal guardrail and structure its textual output. We noted that introducing halting mechanisms such as requesting users inputs in case of missing data (or after a failed calculation) is sufficient in alerting the user of its incomplete output.

We further introduce the DCS publication (Butler & Dressman, 2010) and a follow-up action plan depending on the DCS class outcome as a simple PDF guidelines which are consulted via a RAG step. This rather simple collection of examples serves the purpose of showing the degree of control one can ensure the tool to follow by mixing RAG and prompt.

## A.3   FOLLOWING UP WITH ADDITIONAL TOOLS

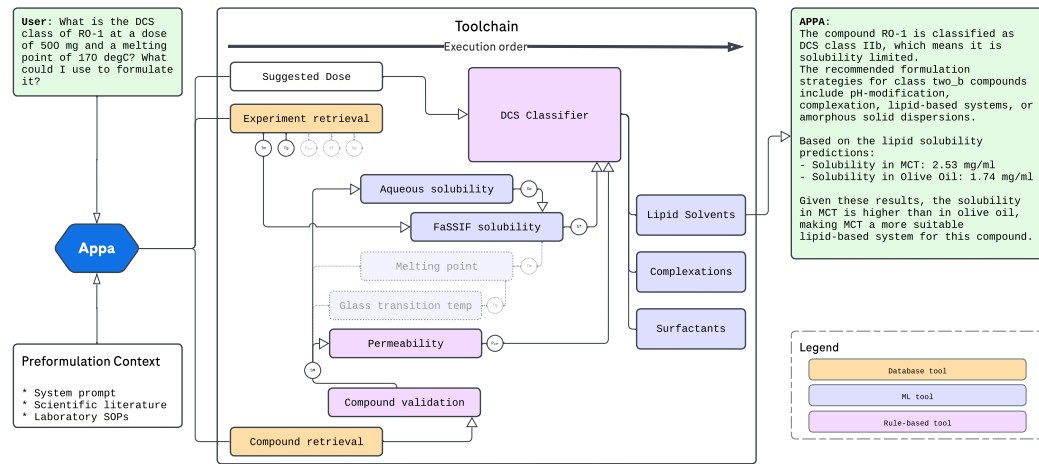

Figure 2: The workflow can be extended with additional steps and tools, depending on the formulation path which is recommended. In this example, after having determined a compound being class IIb, lipid based formulation is identified as a potential way forward. APPA fetches all the models it has available to compare which vehicle is appearing to be best suited to test as additional step.

In a practical scenario, after identifying the probable DCS class of a compound, a formulator would be faced with the task of selecting which vehicles would be best suited to facilitate its development. This entails choosing a class of excipient from the pool of available ones, based on how they modify a compound's solubility in a potential route of administration. APPA automates this step by dynamically selecting the machine learning models for the classes of vehicles which are compatible with the classification's needs, and based on their results, recommends a specific compound choice. This extended workflow is depicted in figure A.3, and showcases a potential multi-step process in the

workflow. We note how our current implementation has currently only a few examples of enabling formulation models, and as a result its presentation serves mostly as a proof of concept.

## A.4 RESULTS : THE COMPLETE CONVERSATION

The examples shown in the result sections focus on the immediate answers offered in response to users' queries. This effectively masks the chain of actions the agent carries out - and how these are finally displayed to the users. We report in this section the complete conversation displayed by APPA in the delivery of the results presented in the main text. We have edited the output by hiding chemical structures in the SMILES string.

```
User: Which one is more soluble in water: RO-1 or RO-2?
```

```
============================= Ai Message =============================
Tool Calls:
  code_to_smiles (call_mabgbd7Cgvf4YiQ74GcamtUH)
  Args:
    code: RO-1
============================= Tool Message =============================
Name: code_to_smiles

SMILESTRING1
============================= Ai Message =============================
Tool Calls:
  is_smiles_valid (call_dyIR78Q57wBkm6gg2kTfxYZE)
 Call ID: call_dyIR78Q57wBkm6gg2kTfxYZE
  Args:
    smiles: SMILESTRING1
============================= Tool Message =============================
Name: is_smiles_valid

true
============================= Ai Message =============================
Tool Calls:
  code_to_smiles (call_h7u570U866LetVaAlWCtthFv)
 Call ID: call_h7u570U866LetVaAlWCtthFv
  Args:
    code: RO-2
============================= Tool Message =============================
Name: code_to_smiles

SMILESTRING2
============================= Ai Message =============================
Tool Calls:
  is_smiles_valid (call_0BC01F6Quj6E9ftUImAorfpP)
 Call ID: call_0BC01F6Quj6E9ftUImAorfpP
  Args:
    smiles: SMILESTRING2
============================= Tool Message =============================
Name: is_smiles_valid

true
============================= Ai Message =============================
Tool Calls:
  calculate_aqueous_solubility (call_50DVqHIf4slKwRBllJ5LSqfy)
 Call ID: call_50DVqHIf4slKwRBllJ5LSqfy
  Args:
    smiles: SMILESTRING1
  calculate_aqueous_solubility (call_Bpf8rRO4Qabx1mfrXDsd38Df)
 Call ID: call_Bpf8rRO4Qabx1mfrXDsd38Df
  Args:
    smiles: SMILESTRING2
============================= Tool Message =============================
Name: calculate_aqueous_solubility

10.349786144139605
============================= Tool Message =============================
Name: calculate_aqueous_solubility

0.6067265461374582
============================= Ai Message =============================

RO-1 is more soluble in water with a solubility of 10.35 mg/ml compared to RO-2 which has a
    solubility of 0.61 mg/ml.
```

```
### Tools Used:
1. **functions.code_to_smiles**: To retrieve the SMILES strings for both compounds.
2. **functions.is_smiles_valid**: To ensure the retrieved SMILES strings are valid.
3. **functions.calculate_aqueous_solubility**: To predict the aqueous solubility of both
     compounds.
```

A more complicated example is to propose APPA a request which is not directly provided by the tools themselves. In this second case we asked it to return the maximal dose in mg a compound can be administered while staying in Class I. To address this request, APPA needs to understand the variables underlying the DCS categorization.

```
    User: Calculate what is the maximal dose in mg that compound RO-3 can be administered
        while still classify as a DCS class 1. Assume a melting temperature of of 170 degC.
        Explain why.
```

```
 =========================== Ai Message ===========================
Tool Calls:
  fetch_exp_data (call_PBQzT8kev2PPkyNVAiTB9iD8)
 Call ID: call_PBQzT8kev2PPkyNVAiTB9iD8
  Args:
    code: RO-3
=========================== Tool Message ===========================
Name: fetch_exp_data

{LIST_OF_VALUES}
=========================== Ai Message ===========================
Tool Calls:
  code_to_smiles (call_X0x47HeD6atH673zhQMQYXdS)
 Call ID: call_X0x47HeD6atH673zhQMQYXdS
  Args:
    code: RO-3
=========================== Tool Message ===========================
Name: code_to_smiles

SMILESTRING
=========================== Ai Message ===========================
Tool Calls:
  is_smiles_valid (call_CVXg2bUj0wrg03F23fLP10yS)
 Call ID: call_CVXg2bUj0wrg03F23fLP10yS
  Args:
    smiles: SMILESTRING
=========================== Tool Message ===========================
Name: is_smiles_valid

true
=========================== Ai Message ===========================
Tool Calls:
  calculate_aqueous_solubility (call_5CxQ3UInFA7MOSFyFvpXiOXK)
 Call ID: call_5CxQ3UInFA7MOSFyFvpXiOXK
  Args:
    smiles: SMILESTRING
=========================== Tool Message ===========================
Name: calculate_aqueous_solubility

0.007315910121087895
=========================== Ai Message ===========================
Tool Calls:
  calculate_fassif_solubility (call_sCUomCidRDyTGSZeGU27a02q)
 Call ID: call_sCUomCidRDyTGSZeGU27a02q
  Args:
    smiles: SMILESTRING
    melting_point: 170
    sol_buffered: 0.007315910121087895
=========================== Tool Message ===========================
Name: calculate_fassif_solubility

0.04648209417216417
=========================== Ai Message ===========================
Tool Calls:
  calculate_peff (call_9SIYRwt4NGhLHsw5B07jUIEA)
 Call ID: call_9SIYRwt4NGhLHsw5B07jUIEA
  Args:
    smiles: SMILESTRING
=========================== Tool Message ===========================
Name: calculate_peff

0.00039492970516104465
=========================== Ai Message ===========================
```

```
Tool Calls:
  get_dcs_class (call_5m8FmlIxdMiW1vsAw6I6Atin)
 Call ID: call_5m8FmlIxdMiW1vsAw6I6Atin
  Args:
    sol_fassif: 0.04648209417216417
    peff: 0.00039492970516104465
    dose: 1
============================= Tool Message =============================
Name: get_dcs_class

{"dcs_class": "one", "dose_number": 0.04302732128617606, "absorption_number":
    4.012169860672085}
============================= Ai Message =============================

The maximal dose in mg that compound RO-3 can be administered while still classifying as a DCS
    class 1 is 23 mg.

Here is the reasoning and tools used:

1. **fetch_exp_data**: Retrieved physicochemical data for RO-1.
2. **code_to_smiles**: Converted the code to a SMILES string.
3. **is_smiles_valid**: Verified the validity of the SMILES string.
4. **calculate_aqueous_solubility**: Predicted the aqueous solubility.
5. **calculate_fassif_solubility**: Predicted the solubility in FaSSIF.
6. **calculate_peff**: Calculated the human effective permeability.
7. **get_dcs_class**: Classified the compound according to the DCS system and determined the
    dose number.

The dose number for a 1 mg dose was found to be 0.043, which means the dose can be increased
    to approximately 23 mg to still be within the DCS class 1 criteria.
```

