# OpenReview forum: "APPA : Agentic Preformulation Pathway Assistant"
_ICLR.cc/2025/Workshop/AgenticAI — ICLR 2025 Workshop AgenticAI Poster_

### Official Review · Reviewer_ZuX4 · 2025-03-02
**APPA: AGENTIC PREFORMULATION PATHWAY ASSISTANT**

**Rating:** 6
**Confidence:** 5

**Review:**

1. Introduces APPA, an LLM-driven agentic framework for preformulation pathway optimization in drug development.

2. Integrates scientific literature, experimental databases, and ML models to assist in drug formulation decisions.

3. Leverages Developability Classification System (DCS) to suggest optimal formulation strategies.

4. Automatically classifies drug candidates and recommends next steps using solubility, permeability, and physicochemical data.

5. Uses retrieval-augmented generation (RAG) and Langchain framework to ensure scientifically grounded outputs.

6. Outperforms standard GPT-4o in drug classification tasks, providing actionable insights beyond retrieval-based LLMs.

7. Supports multi-step query chains, enabling dose optimization, solubility comparisons, and experimental decision guidance.

8. Bridges the gap between manual preformulation workflows and AI-driven automation in pharmaceutical R&D.

9. Evaluation limited to in silico performance—no validation with real-world experimental formulations.

10. Computational efficiency not addressed—unclear impact of APPA’s iterative reasoning on speed and resource usage.

11. No comparison against existing AI-driven formulation tools—unclear how APPA stacks against proprietary pharma solutions.

12. Potential LLM hallucination risks—may generate plausible yet incorrect formulation recommendations.

13. Lacks human-in-the-loop validation—no mechanism to verify APPA’s reasoning against expert formulation scientists.

14. Needs better interpretability in decision-making—unclear why certain formulations are prioritized over others.

15. Promising application but requires further benchmarking, efficiency analysis, and experimental validation.

---

### Official Review · Reviewer_9PCH · 2025-03-02
**hard to follow the result**

**Rating:** 4
**Confidence:** 1

**Review:**

This paper introduces an agentic preformulation pathway assistant (APPA), leveraging large language models coupled to experimental databases and a suite of machine learning models to streamline the preformulation process of drug candidates. APPA integrates domain expertise from scientific publications, databases holding experimental results, and machine learning predictors to reason and propose optimal preformulation strategies based on the current evidence.

The experimental results presented in the paper are hard to follow and judge for non-domain experts. The author should consider to include some qualitative evaluations.

---

### Official Review · Reviewer_XNWC · 2025-03-04
**Is it necessary to rely on LLM to control the workflow in this DCS classification task?**

**Rating:** 5
**Confidence:** 3

**Review:**

**Paper Summary:**

This paper introduces an agent system designed to assist in the preformulation process. The system responds to user queries by retrieving experimental data and utilizing existing machine learning models to predict various physicochemical properties and assay outcomes, thereby facilitating efficient and informed decision-making during the preformulation stage.

**Strengths:**

The proposed agent system has the potential to accelerate the preformulation process.

**Weaknesses:**

- **W1:** The experimental validation relies on only four samples, which is insufficient for a comprehensive evaluation of the framework's efficacy. To ensure robustness and reliability, it is recommended to test at least 500 samples.

- **W2:** The claim that the agent system suggests meaningful next steps in formulating compounds lacks quantitative evidence. Providing detailed, quantitative results would help substantiate the system's effectiveness in this regard.

- **W3:** The paper does not offer a comparison between the proposed agent framework and existing model-based DCS classification methods[1], such as those discussed by Lange et al. (2024b). A comparison would provide valuable context on the performance and advantages of the new framework.

- **W4:** The authors should provide a clearer rationale for opting for an LLM-based agent system over a traditional system. Given the simplicity of the workflow illustrated in Figure 1 and the limited number of tools involved, it is unclear why a traditional system would not suffice. In other words, is it necessary to rely on LLM to control the workflow in this DCS classification task? This decision should be justified with experimental evidence to support the choice of an LLM-based agent framework.



[1] Justus Johann Lange, Andrea Anelli, Jochem Alsenz, Martin Kuentz, Patrick J O’Dwyer, WiebkeSaal, Nicole Wyttenbach, and Brendan T Griffin. Comparative analysis of chemical descriptorsby machine learning reveals atomistic insights into solute–lipid interactions. Molecular Pharma-ceutics, 2024b.

---

### Decision · Program_Chairs · 2025-03-05

**Decision:**

Accept (Poster)

**Comment:**

The authors have submitted a revised version accommodating reviewer comments. Based on the second round of reviews, the decision has been made to accept the paper for poster presentation.